# Evaluating Leaf Wettability and Salt Hygroscopicity as Drivers for Foliar Absorption

**DOI:** 10.3390/plants12122357

**Published:** 2023-06-18

**Authors:** Neriman Tuba Barlas, Héctor Alejandro Bahamonde, Carlos Pimentel, Pedro Domínguez-Huidobro, Carlos M. Pina, Victoria Fernández

**Affiliations:** 1Department of Soil Science and Plant Nutrition, Faculty of Agriculture, Ege University, 35100 Izmir, Türkiye; tuba.barlas@ege.edu.tr; 2Facultad de Ciencias Agrarias y Forestales, Universidad Nacional de La Plata, Diagonal 113 No. 469, La Plata 1900, Argentina; bahamondehector2019@gmail.com; 3Univ. Grenoble Alpes, Univ. Savoie Mont Blanc, CNRS, IRD, Univ. Gustave Eiffel, ISTerre, 38000 Grenoble, France; carlos.pimentel-guerra@univ-grenoble-alpes.fr; 4Departamento de Sistemas y Recursos Naturales, Universidad Politécnica de Madrid, CC/José Antonio Novais 10, 28040 Madrid, Spain; pedro.dominguezhuido@alumnos.upm.es; 5Departamento de Mineralogía y Petrología, Facultad de Ciencias Geológicas, Universidad Complutense de Madrid, C/José Antonio Novais, 12, 28040 Madrid, Spain; cmpina@geo.ucm.es; 6Instituto de Geociencias IGEO (UCM—CSIC), 28040 Madrid, Spain; 7Centro para la Conservación de la Biodiversidad y el Desarrollo Sostenible, E.T.S.I. Montes, Forestal y del Medio Natural, Universidad Politécnica de Madrid, 28040 Madrid, Spain

**Keywords:** crops, hygroscopic salts, foliar uptake, leaf anions, leaf wetting

## Abstract

The objective of this study was to evaluate the rate of foliar absorption of magnesium (Mg) salts with different deliquescence and efflorescence relative humidity values (DRH and ERH, also known as point of deliquescence (POD) and point of efflorescence (POE), respectively) when supplied to leaves of model plants with different wettability properties. For this purpose, a greenhouse pot experiment was conducted with lettuce (very wettable), broccoli (highly unwettable) and leek (highly unwettable). Foliar sprays contained 0.1% surfactant plus 100 mM Mg supplied as MgCl_2_·6H_2_O, Mg(NO_3_)_2_·6H_2_O or MgSO_4_·7H_2_O. Leaf Mg concentrations were determined 1 and 7 days after foliar application. Anion concentrations were also measured in lettuce where a significant foliar Mg absorption was detected. Leaf wettability, leaf surface free energy and fertilizer drop deposit appearance onto the foliage were assessed. It is concluded that despite including a surfactant in the spray formulation, leaf wettability plays a major role in foliar Mg absorption.

## 1. Introduction

Magnesium is an essential element for plants, and Mg^2+^ is known as the second most abundant plant cation [1]. The leaf concentration of Mg often varies between 0.1 and 1.00%, but the sufficiency level for many species is in the range of 0.1 to 0.3% [2]. In addition, it is reported that plant Mg contents can vary considerably according to different species [2]. This element is essential for photosynthesis [3], and it is part of the structure of chlorophyll (15 to 20% of total leaf Mg may be linked with chlorophyll pigments [4]). Magnesium is also important for facilitating phosphorus transport [5,6,7]. Moreover, Mg plays a key role in plants as it acts as a cofactor of enzymes involved in photosynthetic carbon fixation, plant metabolism [8,9,10] and other vital functions such as protein synthesis, sugar transport, energy metabolism, nitrogen use and stress tolerance mechanisms [4,11,12,13,14].

Due to the wide range of functions of Mg, plant growth and physiology are significantly limited under magnesium deficiency conditions as observed in many areas of the world [6]. The Mg content of plants grown under sufficient Mg availability generally varies from 0.15 to 0.35% [6]. Since it is a mobile element, deficiency symptoms are primarily seen on older leaves which develop interveinal chlorosis and accumulate sugars [6,15,16].

Foliar sprays are often used in commercial agriculture to complement root treatments, or as an alternative means to supply nutrients to the plants under conditions limiting root absorption and/or plant transport, or in cases when the nutrient demand exceeds the root absorbing/plant transport capacity [17,18]. The process of foliar nutrient uptake is complex, and many factors may influence plant responses to foliar sprays such as the prevailing environmental conditions during treatment, the physicochemical properties of active and spray formulation ingredients or plant physiological and anatomical features [19]. Leaf surface wettability is an important factor affecting the response of plants to agrochemical spray treatments which can be characterized by measuring contact angles [18]. Surfaces that are wettable will have a high area of contact between spray drops and the treated leaf surfaces hence increasing the potential for foliar absorption to occur [18]. By contrast, very unwettable leaf surfaces may even have drop repellence and a reduced contact area of spray drops with the treated surfaces [18]. Factors such as drought, relative humidity (RH), irradiation, nutrient deficiencies or microbial colonization can change leaf surface properties and have an effect on leaf wettability [18,19]. Despite leaf wettability being an important preliminary variable affecting foliar absorption, transport across the leaf epidermis may also be limited by the physico-chemical nature of the cuticle and cell wall [18]. Various studies carried out with different crop species reported positive plant responses and increased tissue Mg values in association with the application of foliar Mg sprays, e.g., [20,21,22,23,24].

The importance of RH as a key factor affecting foliar absorption and aerosol effects on the foliage of plants has been emphasized in previous investigations [25,26,27,28]. Some studies analyzed the performance of hygroscopic salts as aerosol particles in relation to foliar absorption, e.g., [28,29], and stomatal penetration [30], focusing on the role of salt deliquescence relative humidity (DRH) or point of deliquescence (POD) [25,28,30]. This parameter significantly varies with temperature and refers to the RH at which an initially crystalline compound dissolves after water sorption from the surrounding environment [31]. This implies that after the foliar application of a certain chemical, salt spray deposits may rehydrate and become liquid if, for a given temperature, RH reaches or goes above the DRH of such chemicals. Compounds with lower DRH values have been generally assumed to be more favorable to be taken up by the foliage after foliar treatment [28]. 

On the other hand, salt efflorescence relative humidity (ERH) or point of efflorescence (POE) refers to the RH at which a concentrated solution begins to crystallize [31]. This parameter has deserved limited attention regarding the absorption of foliar sprays and has only been reported for a few calcium (Ca) and potassium (K) salts [31,32]. Efflorescence RH does not seem to vary so much with temperature as DRH [31,32]. As foliar sprays are applied as liquids, drop drying and salt crystallization will be determined by the ERH, while salt hydration at an increased ambient RH (e.g., during the night) will be associated with the DRH of the compound/s under the prevailing temperature conditions [31,32]. 

Provided the assumed significance of the DRH and ERH of agrochemical sprays as described above, we aimed at assessing the absorption of foliar-applied Mg supplied as salts with markedly different hygroscopic properties (i.e., magnesium nitrate, magnesium sulfate and magnesium chloride), by treating three plant species with extreme leaf wettability properties (i.e., highly wettable lettuce, and extremely unwettable broccoli and leek). By determining leaf Mg concentrations 1 and 7 days after foliar application, the hypotheses tested were: (i) the salt with the lowest ERH will lead to the highest Mg foliar uptake rates at least 1 day after foliar treatment, (ii) the salt with the lowest DRH will lead to the highest foliar Mg absorption 7 days after foliar spraying, and (iii) despite adding a surfactant to the foliar Mg solutions which reduced surface tension to 27 mN m^−1^, the intrinsic leaf wettability characteristics of each species will still influence the rate of foliar Mg absorption.

## 2. Results

### 2.1. Magnesium Salt Physico-Chemical Properties

The properties of the Mg salts applied as foliar sprays are shown in Table 1. Magnesium cation was always supplied at a 100 mM concentration. The most soluble and hygroscopic compound is MgCl_2_ 6H_2_O, followed by Mg(NO_3_)_2_ 6H_2_O and MgSO_4_ 7H_2_O. The hygroscopicity of the salts was measured at 20 °C, and the DRH varied between 40 and 92 RH%, while the ERH varied between 15 and 83 RH%.

### 2.2. Leaf Surface Structure, Wettability and Surface Free Energy Related Parameters

The leaf surfaces evaluated had contrasting wettability characteristics, with lettuce being quite wettable compared to leek and broccoli, which were extremely unwettable (Table 2). Indeed, the upper (adaxial) and lower (abaxial) side of leek leaves and the lower side of broccoli leaves proved to be water repellent, while the upper side of broccoli had a rose petal effect (i.e., high contact angles but drop adherence; [35]). By contrast, lettuce leaf surfaces were wettable (*θ* < 90 °C) for all the measuring liquids, with the lower leaf side having low water contact angles of 58° (Table 2). 

The total surface free energy of lettuce leaf surfaces was high and ranged from 36 to 40 mJ m^−2^, also having a high polarity (values of 22 to 24% (Table 3). By contrast, both leek surfaces and the abaxial side of broccoli leaves had the lowest total surface free energy values owing to the occurrence of apolar waxes providing a complex surface nano-topography. The polarity and total surface free energy of the upper leaf side of broccoli was higher, however, despite the measured high water contact angles, and this is a feature associated with the rose petal effect observed for this material. The solubility parameter (*δ*) of leek and broccoli leaf surfaces was low due to the occurrence of nano-structured epicuticular waxes which make the surfaces rather apolar and extremely rough [36,37].

### 2.3. Foliar Mg Trial

#### 2.3.1. Environmental Conditions

The temperature (°C) and RH (%) conditions during the experimental period are shown in Figure 1. In the first few days, temperatures ranged from ~20 °C to ~10 °C. This temperature difference was, however, decreased by ~5 °C in the last days of the experiment. Attenuation was also observed in the difference between the minimum and maximum RH values. In the first few days, the RH was around 60%, with minimum values of 30–40% when the maximum temperature was reached. In the last few days, there was almost no difference in the RH values measured throughout the day. This implies that, after the application of the foliar Mg sprays, magnesium chloride could remain as a (concentrated) solution throughout the whole experimental period, as the RH was always above the ERH of this salt (15% at 20 °C). In the case of magnesium nitrate, drop deposits may have experienced hydration–dehydration cycles, as there were days when the RH reached values below the ERH, and also above the DRH. In the case of magnesium sulfate, the RH in the greenhouse was always below the ERH, which implies that fertilizer drops dried out and crystallized shortly after foliar treatment and did not liquefy again, as RH never exceeded its DRH value.

#### 2.3.2. Salt Deposition after Foliar Mg Application

The appearance of Mg salt deposits after foliar application onto the foliage of lettuce, broccoli and leek was analyzed by SEM. Only the results associated with the salts showing the highest (magnesium sulfate) and lowest (magnesium chloride) DHR and ERH values are shown as an example (Figure 2). The typical distribution morphology of the salt deposits formed during droplet drying on the leaves can be observed in Figure 2. In the case of lettuce (Figure 2A–C), both magnesium chloride (Figure 2B) and magnesium sulfate (Figure 2C) appear to form a quite homogeneous film covering the surface of the leaves, as evidenced by the change in the appearance of the stomata when compared to stomata on an untreated leaf (Figure 2A).

In the case of broccoli (Figure 2E–G), only magnesium chloride forms a layer on the leaf surface, which clearly visible forms patches and covers the waxes (Figure 2E). It is also possible to see some small idiomorphic crystals, which seem to be hydrated magnesium chloride. When magnesium sulfate was supplied to the foliage, the waxes in the broccoli surface seemed to be clustered by the precipitate formed (Figure 2F). In leek (Figure 2H–J), the formation of precipitates was only observed when magnesium chloride applied was sprayed. In this case, on the ridges of the leek, a thin layer can be seen covering the waxes (Figure 2H). The surface of the leek leaf after treatment with magnesium sulfate resembled that of the untreated leaf (Figure 2H,J).

#### 2.3.3. Leaf Mg Concentrations after Foliar Treatments

The rates of foliar Mg absorption of lettuce, broccoli and leek plants are shown in Figure 3, Figure 4 and Figure 5, lettuce being the most permeable to the Mg sprays.

The effect of all foliar Mg treatments in increasing lettuce leaf Mg concentrations was statistically significant at both sampling times (1 and 7 days after Mg spraying). One day after foliar treatment, the highest leaf Mg concentrations were recorded for magnesium sulfate, followed by magnesium nitrate and magnesium chloride (Figure 3). For both sampling dates, the tissue Mg values recorded for foliar-treated lettuce plants were significantly higher than those of untreated ones. In general, 7 days after foliar treatment, the highest leaf Mg concentrations were recorded after magnesium chloride and magnesium nitrate spraying.

In broccoli, the highest leaf Mg concentrations 1 day after foliar Mg supply were measured after magnesium nitrate spraying, but the values were not significantly different from the rest of the treatments. However, 7 days after foliar Mg application, a significant leaf Mg concentration increase was recorded after magnesium chloride supply (Figure 4).

Concerning the rate of Mg absorption by leek leaves, no significant differences were determined either 1 or 7 days after foliar Mg spraying. The highest tissue Mg concentrations were always measured after supplying magnesium chloride, but the values were not significantly different between treatments (Figure 5).

#### 2.3.4. Lettuce Leaf Anion Concentrations after Foliar Treatment

Since significant Mg concentration differences between Mg salt treatments and sampling dates were only recorded for wettable lettuce leaves, leaf-accompanying anion concentrations were only determined for this species, as shown in Figure 6.

Magnesium chloride application increased leaf chloride concentration chiefly 1 day after treatment, but the values were not statistically significant from those of untreated plants, also after 1 week (Figure 6A). Leaf nitrate concentrations of magnesium-nitrate-sprayed lettuce leaves were extremely high 1 day after application, but markedly decreased to values similar to those recorded in the rest of the treatments 7 days after foliar spraying (Figure 6B). Evidence for sulfate foliar absorption after magnesium sulfate spraying was gained from the high leaf tissue values determined. The sulfate values recorded for the leaves collected 1 and 7 days after foliar spraying were actually not significantly different (Figure 6C).

## 3. Discussion

Foliar sprays are often applied to crop species in many areas of the world [20,21,22,23,24], but the response of plants to the treatments is often variable as described for other elements [17,19]. Hence, specific trials are required for attempting to clarify which are the major factors influencing the absorption of foliar agrochemical treatments as a prerequisite for optimizing the efficacy of foliar sprays in the future. For this purpose, working with Mg as a model plant nutrient, we evaluated the influence of leaf wettability and salt ERH and DRH as key factors affecting the absorption of foliar-applied Mg solutions.

Leaf wettability generally refers to the interaction of leaf surfaces with water drops [18,38]. In this study, we applied a surfactant to lower Mg solution surface tension to approximately 27 mN m^−1^ and avoided spray drop repulsion in the lower side of broccoli and both sides of leek leaves, hence giving the chance for Mg uptake to occur after foliar application [18]. We selected a concentration of 100 mM Mg because it has been suggested to be optimal for many cultivated plants, also considering the supply of magnesium chloride, magnesium nitrate and magnesium sulfate foliar fertilizers in several studies, e.g., [39,40,41,42,43,44].

Relative humidity has been mentioned as a key factor affecting foliar nutrient absorption, and major importance has been chiefly attributed to salt DRH, e.g., [28], an idea that strongly grew among the technical and scientific community during recent decades. Aware of the significance of salt ERH as described in recent investigations [31,32], we aimed at testing which of these parameters may be more influential for foliar Mg uptake. Aware of the major impact of leaf wettability, at least for surface interactions between spray drops/deposited particles and leaf surfaces, we chose lettuce as a highly wettable leaf, and broccoli and leek leaves as examples of highly unwettable surfaces with water drop repulsion and/or rose petal effect [18,35].

Should DRH be a major factor as hypothesized in this study, we must record Mg tissue increases several days after spraying foliar magnesium chloride (41% DRH) and to a lower extent, magnesium nitrate (62% DRH) solutions, when hydration-dehydration cycles may have occurred on spray drop deposits as affected by diurnal, greenhouse temperature and RH fluctuations (Figure 1). This hypothesis was not clearly verified, because in lettuce, the highest tissue Mg increases after 7 days were recorded after foliar magnesium nitrate application, followed by magnesium chloride which has a lower DRH. In the case of leek and chiefly broccoli where differences were significant, the highest tissue Mg values were determined after foliar magnesium chloride spraying, magnesium nitrate and magnesium sulfate leading to similar leaf Mg concentrations. Hence, it could not be concluded that the compound with the lowest DRH led to the highest leaf Mg increments after foliar application. However, a trend for increased magnesium chloride absorption was observed for wettable lettuce and extremely unwettable broccoli and leek leaves.

Regarding our hypothesis that the Mg salt with the lowest ERH will lead to the highest Mg foliar uptake rates 1 day after foliar treatment, which was magnesium chloride, again no conclusive results were gained as described above for Mg salt DRH. The highest rate of Mg absorption of wettable lettuce leaves was recorded after foliar magnesium sulfate application, which is the compound with the highest ERH value of the ones tested. Only in leek, the highest absorption of Mg was measured 1 day after foliar magnesium chloride application, which, however, was not significant compared to the rest of the treatments. This again indicates that no clear trend could be established between the salt ERH and the Mg foliar uptake of lettuce, leek and broccoli plants. 

For crystallization, drops of the sprayed Mg salts have to lose almost all of their water content during drying, so that super-saturation with respect to solid phases (magnesium chlorides, nitrates and sulfates with different numbers of water molecules in their chemical formulae) is achieved. If we assume that at the ERH, solution drops are saturated for MgCl_2_·6H_2_O, Mg(NO_3_)_2_·6H_2_O or MgSO_4_·7H_2_O, the corresponding Mg concentrations will be 7.44, 6.8 and 3.89 M, respectively (Table 1). However, as stated above, only magnesium sulfate reached the ERH during the first desiccation cycle. In the case of magnesium nitrate, such concentrations could be reached numerous times during the experimental period, because several hydration–drying cycles may have occurred due to the minimum and maximum RH values measured in the greenhouse (which were below and above ERH and DRH, respectively). Similarly, it can also be derived that the lack of significant foliar Mg absorption in the broccoli and leek leaves is not related to a low Mg concentration in the foliar sprays but is rather associated with other factors affecting the process of foliar uptake.

As a third hypothesis, we established that despite adding a powerful surfactant to the Mg solutions, the intrinsic leaf wettability characteristics of each species will still influence the rate of foliar Mg absorption. Our results show that wettable lettuce leaves were the most permeable, with broccoli and leek leaf surface permeability being low for all the Mg salt plus surfactant solutions tested. It seems hence plausible that leaf wettability may be a major factor affecting the absorption of foliar sprays which overrules the influence of other physico-chemical factors such as salt DRH and ERH, as shown in our study. We are, however, aware that in addition to the initial influence of leaf surface–drop interactions, the permeability of plant cuticles to solutes/solvents is influenced by the solubility and diffusivity of the applied chemicals [36]. While diffusivity is a kinetic parameter mainly associated with the molecular size of a compound in relation to the structure of the matrix, solubility is a thermodynamic parameter associated with the affinity of a certain chemical for the cuticle/cell wall [36]. These factors together with the potential contribution of stomata to the foliar absorption process [45,46] have not been evaluated in this study and will require further investigation, also regarding variations between species and other abiotic/biotic variables. 

Leaf anion concentrations are comparatively less often determined in leaf tissues, but also important for plant physiology and metabolism [32,47]. We only analyzed the anion concentrations of lettuce leaves because of the high Mg absorption rates recorded. We observed that the nitrate and sulfate supply led to significant tissue nitrate and sulfate increases at least 1 day after foliar application. However, the leaf chloride concentrations determined after magnesium chloride foliar application were only different from the rest of the treatments 7 days after foliar spraying. This suggests that chloride may have been absorbed over time after foliar treatment. Nonetheless, leaf chloride values generally decreased 1 week after foliar spraying which may be due to plant physiological and metabolic changes. By contrast, magnesium sulfate foliar application led to a major tissue sulfate increase which remained high 7 days after treatment. This may be related to the limited mobility of foliar-absorbed sulfate and to factors related to the sulfur transport and metabolism in the plant [6,48].

## 4. Conclusions

In this experiment, we assessed the rate of Mg uptake in lettuce, leek and broccoli by applying solutions of Mg salts with different DRH and ERH values. We could only detect significant rates of Mg absorption in lettuce both 1 and 7 days after treatment, regardless of the Mg salt hygroscopicity characteristics.

Despite having included a surfactant for reducing the surface tension of water (approximately 72 mN m^−1^) to 27 mN m^−1^ in the Mg spray solutions, significant Mg uptake was only recorded for highly wettable lettuce, versus the generally low or negligible rate of Mg foliar absorption of extremely unwettable broccoli and leek leaves. No clear influence of Mg salt hygroscopicity was observed, hence future foliar absorption studies should not only focus on testing plant responses to nutrient compounds based on selecting salts with low points of deliquescence. It is concluded that the influence of apparently positive physico-chemical properties of agrochemical formulations cannot be simply predicted in terms of increased foliar absorption rates. For improving the performance of foliar nutrient sprays, it will be necessary to carry out holistic studies, considering factors such as leaf wettability, plant surface structure and composition, the contribution of leaf surface structures to foliar absorption, the potential response of epidermal cell ion transporters to foliar treatments or major physico-chemical properties of spray active ingredients and adjuvants. 

## 5. Materials and Methods

### 5.1. Plant Material

Lettuce (*Lactuca sativa* var. Romana), broccoli (*Brassica oleracea* var. Trajano F1) and leek (*Allium porrum* var. Helvetia) were used as model species with quite wettable (lettuce) and extremely unwettable (broccoli and leek) leaves. Approximately 1-month-old seedlings were acquired from Semilleros El Mirador (El Mirador, Murcia, Spain) and were cultivated under optimal nutrition and irrigation conditions for 2 months in a greenhouse (School of Forest Engineering, Universidad Politécnica de Madrid, Spain).

### 5.2. Chemicals and Mg Salt Properties 

The compounds supplied as foliar sprays were magnesium chloride (MgCl_2_·6H_2_O), magnesium nitrate (Mg(NO_3_)_2_·6H_2_O) and magnesium sulfate (MgSO_4_·7H_2_O) sources (all of them were ACS reagents, 99%; Sigma–Aldrich, Taufkirchen, Germany). All treatments contained 0.1% Genapol X-80 (Sigma–Aldrich) which reduced solution surface tension to approximately 27.21 ± 0.66 mN m^−1^ [32]. The solubility in molarity (M) of these salts was determined by PHREEQC code [33] using the Pitzer database. The values of Mg(NO_3_)_2_·6H_2_O are not included in this database; therefore, the saturation molality (m) of this salt was taken from [34] for calculating its solubility (molarity, M).

### 5.3. Deliquescence and Efflorescence Relative Humidity of Mg Salts

The DRH (or POD) and ERH (or POE) of the Mg salts were assessed using a climatic chamber (MKF 56, Binder, Tuttlingen, Germany) under controlled relative humidity (RH) and temperature conditions. Two grams of each Mg salt were placed in a Petri dish which was kept open in the climatic chamber at a fixed temperature of 20 °C. The R.H. was gradually raised from 20% to 98% at different steps (5% every half an hour) to observe the process of hydration with an optical microscope (DM300, Leica Microsystems, Mannheim, Germany). When becoming close to the DRH of the salts (crystals began to lose their shape and become round), the RH was maintained for a longer period of time to clearly see them becoming liquid. After having identified the DRH, the Mg salts were kept at 98% RH overnight, for subsequently evaluating the process of water desorption. Then, the ERH was determined during the next day by gradually decreasing the RH until the formation of crystals in the concentrated solutions was observed [31,32].

In addition to climate chamber trials, the Mg salt EDH and DHR were estimated at 20 °C by dynamic vapor sorption (DVS) using a TGA Q5000 instrument (TA instruments, New Castle, DE, USA; Thermal Analysis and Calorimetry Service IQAC-CSIC, Barcelona Spain). For this purpose, approximately 10 mg of the initial salt was placed in the TGA. Prior to the sorption–desorption experiment, salts were kept at 20 °C and 0% RH for 300 min. Afterwards, RH was increased by 0.1%/min up to 95% RH for estimating the DRH, and after reaching the maximum RH, it was gradually decreased until 0% RH at a rate of 0.1%/min to determine the ERH.

### 5.4. Foliar Mg Absorption Trial

Approximately 3-month-old lettuce, broccoli and leek plants grown in the greenhouse (School of Forest Engineering, Technical University of Madrid, Spain), were sprayed with Mg solutions including a surfactant. Leaves of these plants were collected 1 and 7 days after treatment for evaluating the rate of foliar uptake of Mg. The experiment was organized following a completely randomized experimental block design with 4 replications per treatment. During the trial development, the environmental conditions in the greenhouse were recorded with a HOBO MX1101 sensor (Onset, Bourne, MA, USA).

Treatment solutions contained 100 mM Mg, supplied as magnesium chloride, magnesium nitrate or magnesium sulfate plus 0.1% Genapol X-80 surfactant. Root Mg absorption following Mg spray application was prevented by covering the pot substrates with aluminum foil. Foliar sprays were supplied between 8 and 9 a.m. to benefit from stomatal opening. At the time of Mg treatment, the plants were healthy, well-nourished and adequately irrigated. 

At the time of harvesting, leaves were excised and thoroughly washed in an acidulated 0.1% detergent solution by scrubbing the surface with the fingers. They were subsequently rinsed with abundant tap water followed by distilled water. Clean leaves were dried in an oven at 70 °C for 2 days and then ground for element analysis. Magnesium was determined by inductively coupled plasma (ICP, Perkin-Elmer, Optima 3000; CEBAS-CSIC Analysis Service, Murcia, Spain). Additionally, after gaining evidence for foliar Mg absorption in some species, concentrations of the anions chloride (Cl^−^), nitrate (NO_3_^−^), and sulfate (SO_4_^2−^) were analyzed via liquid chromatography (CEBAS-CSIC Analysis Service, Murcia, Spain) after the extraction of approx. 0.3 g dry weight (DW) of leaf tissue in distilled water for 2 h, centrifugation and filtration with 0.45 µm filters.

### 5.5. Scanning Electron Microscopy

The distribution and composition of Mg salt deposits onto the foliage of Mg sprayed leaves were analyzed using field emission scanning electron microscopy (SEM, SIGMA 300 VP, Zeiss, Germany, coupled to an X-ray, EDX detector, Universidad Miguel Hernández, Elche, Spain). Fresh leaf sections of approximately 0.5 cm^2^ were cut, and surfaces were directly observed without sputtering at 10.0 kV and an approximately 8.5 mm working distance.

### 5.6. Contact Angles and Surface Free Energy

The contact angles of approximately 2 µL drops of distilled water, glycerol (ReagentPlus, 99%, Sigma Aldrich, St. Louis, MO, USA) and diiodomethane (ReagentPlus, 99%, Sigma Aldrich) deposited onto adaxial and abaxial surfaces of each species were measured at room temperature with a Drop Shape Analysis System (DSA 100, Krüss, Hamburg, Germany; 1 mL syringe with a 0.5 mm diameter needle). Contact angles were automatically calculated by fitting a side-view image of the captured drop to a curve calculated using the Tangent equation [49]. 

For estimating the total surface free energy (*γ*), its components (i.e., the Lifshitz–van der Waals (*γ*_s_^LW^) and acid-base (*γ*_s_^AB^; *γ*^+^ and *γ*_s_^−^)) and the solubility parameter (*δ*) of the different leaf surfaces [36], the surface tension values of 3 liquids were taken into account: water (*γ*_l_ = 72.80 mJ m^−2^, *γ*_l_^LW^ = 21.80 mJ m^−2^, *γ*_l_^+^ = *γ*_l_^−^ = 25.50 mJ m^−2^), glycerol (*γ*_l_ = 63.70 mJ m^−2^, *γ*_l_^LW^ = 33.63 mJ m^−2^, *γ*_l_^+^ = 8.41 mJ m^−2^, *γ*_l_^−^ = 31.16 mJ m^−2^) and diiodomethane (*γ*_l_ = *γ*_l_^LW^ = 50.80 mJ m^−2^, *γ*_l_^+^ = 0.56 mJ m^−2^, *γ*_l_^−^ = 0 mJ m^−2^) [37].

### 5.7. Data Analysis

Exploratory data analyses were carried out to check for compliance with the assumptions of homoscedasticity and data normality. To verify the homoscedasticity of data variances, Bartlett’s tests were performed. When variances were not homogeneous, data transformations were performed. The normality of the data was checked with the Kolmogorov–Smirnov (KS) test.

Differences in contact angles, cations and anion concentrations in leaves were estimated by performing two-way ANOVA analyses. Tukey’s honest significance tests (HDS) were carried out for estimating differences between factors when F-values were significant (*p* < 0.05). Statistical analyses were carried out using SPSS 15.0 (SPSS Inc., Chicago, IL, USA).

## Figures and Tables

**Figure 1 plants-12-02357-f001:**
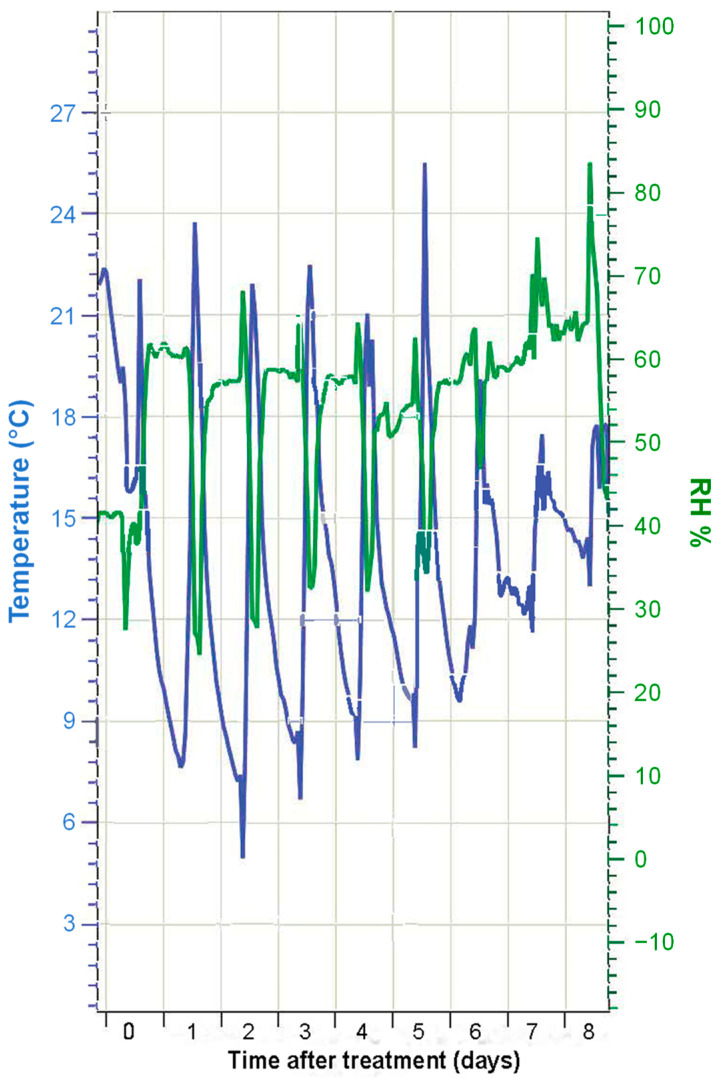
Air temperature (°C) and relative humidity (RH, %) values recorded at the greenhouse during the experiment.

**Figure 2 plants-12-02357-f002:**
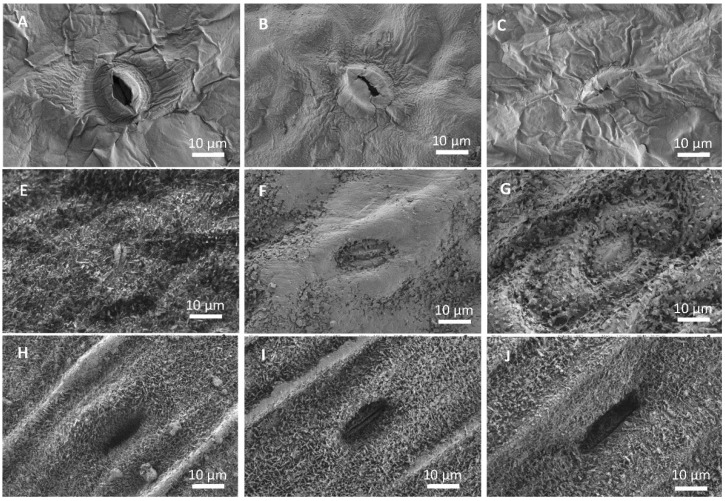
Scanning electron micrographs of the adaxial side of untreated (**A**,**E**,**H**), magnesium chloride (**B**,**F**,**I**) and magnesium sulfate (**C**,**G**,**J**) lettuce (**A**–**C**), broccoli (**E**,**F**) and leek (**H**–**J**) leaves, 7 days after foliar spraying.

**Figure 3 plants-12-02357-f003:**
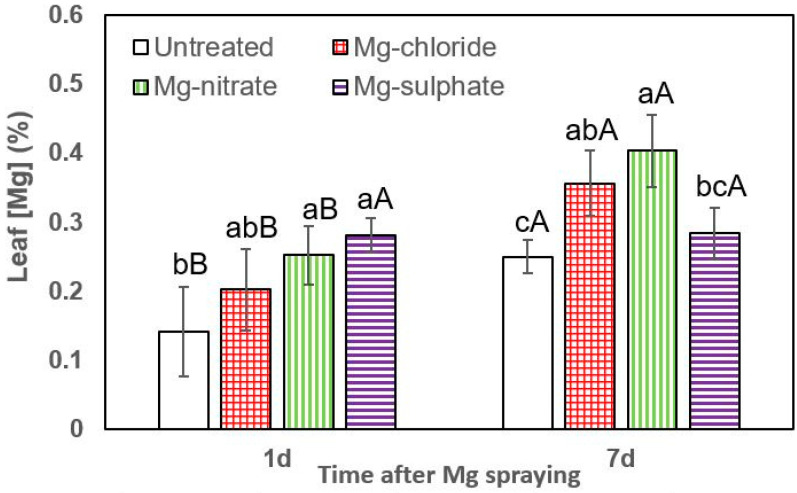
Leaf Mg concentrations of lettuce 1 and 7 days (d) after foliar application of magnesium chloride, magnesium nitrate and magnesium sulfate compared to untreated plants. Data are means ± SD (n = 4). For the same sampling date, values marked with different lower-case letters are significantly different according to Tukey´s HDS test (*p* ≤ 0.05). For the same foliar Mg treatment, capital letters indicate significant differences between sampling dates.

**Figure 4 plants-12-02357-f004:**
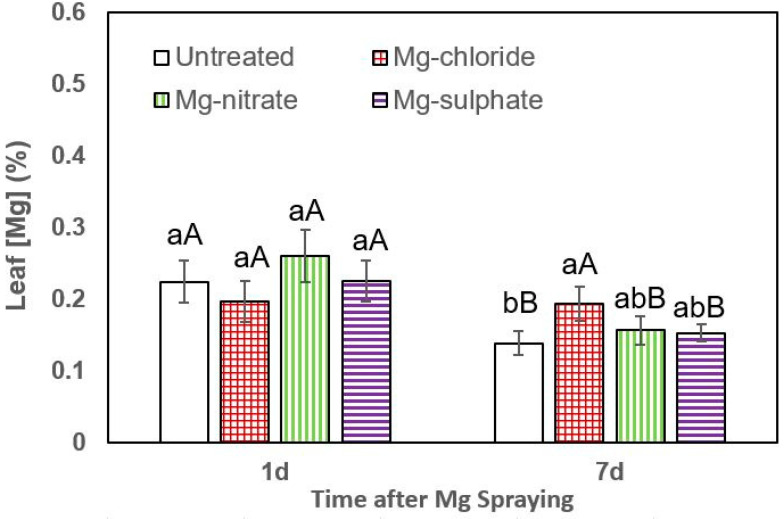
Leaf Mg concentrations of broccoli 1 and 7 days (d) after foliar application of magnesium chloride, magnesium nitrate and magnesium sulfate compared to untreated plants. Data are means ± SD (n = 4). For the same sampling date, values marked with different lower-case letters are significantly different according to Tukey´s HDS test (*p* ≤ 0.05). For the same foliar Mg treatment, capital letters indicate significant differences between sampling dates.

**Figure 5 plants-12-02357-f005:**
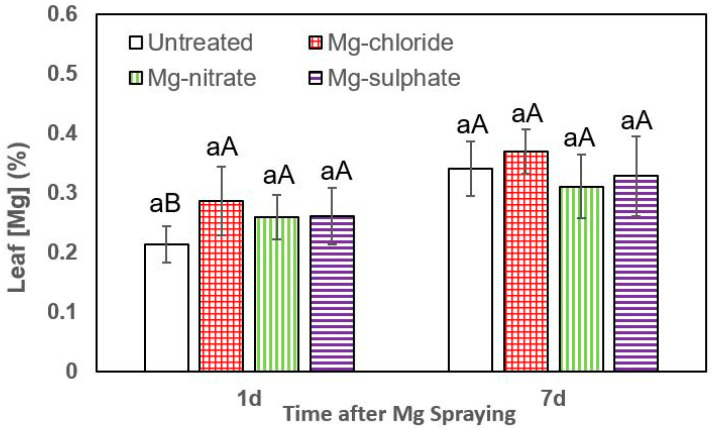
Leaf Mg concentrations of leek 1 and 7 days (d) after foliar application of magnesium chloride, magnesium nitrate and magnesium sulfate compared to untreated plants. Data are means ± SD (n = 4). For the same sampling date, values marked with different lower-case letters are significantly different according to Tukey´s HDS test (*p* ≤ 0.05). For the same foliar Mg treatment, capital letters indicate significant differences between sampling dates.

**Figure 6 plants-12-02357-f006:**
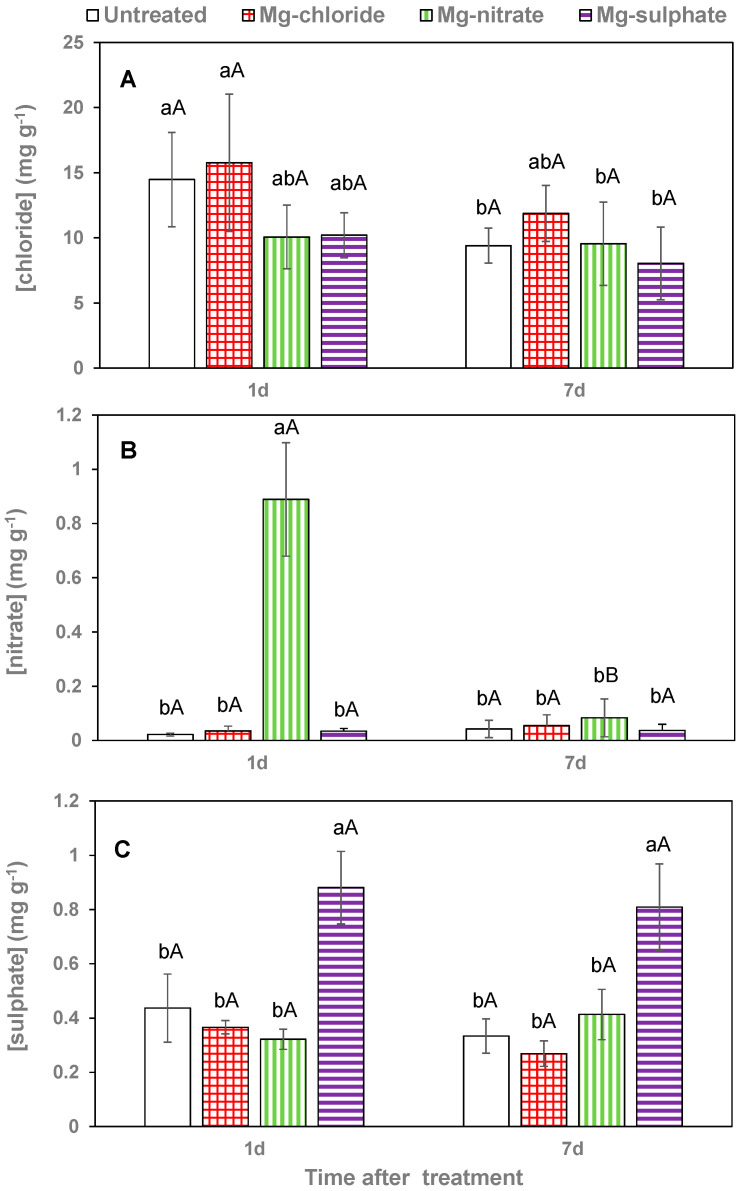
Leaf chloride (**A**), nitrate (**B**) and sulfate (**C**) concentrations 1 and 7 days (d) after foliar application of magnesium chloride, magnesium nitrate and magnesium sulfate compared to untreated lettuce plants. Data are means ± SD (n = 4). For the same sampling date, values marked with different letters are significantly different according to Tukey´s HDS test (*p* ≤ 0.05). For the same foliar Mg treatment, capital letters indicate significant differences between sampling dates.

**Table 1 plants-12-02357-t001:** Solubility, Mg and anion concentrations, approximate deliquescence (DRH or POD) and efflorescence relative humidity (ERH or POE) at 20 °C.

Compound	Solubility (M)	[Mg] (mM)	[anion] (mM)	DRH (%)	ERH (%)
MgCl_2_·6H_2_O	7.74 ^1^	100	200	41	15
Mg(NO_3_)_2_·6H_2_O	6.80 ^2^	100	200	62	32
MgSO_4_·7H_2_O	3.89 ^1^	100	100	92	83

^1^ Calculated using Pitzer.dat [33], ^2^ Calculated using Pitzer.dat [33] with the saturation molalities from [34].

**Table 2 plants-12-02357-t002:** Contact angles of water (*θ_w_*), glycerol (*θ_g_*) and diodomethane (*θ_d_*) with adaxial and abaxial leaf surfaces of lettuce, broccoli and leek plants. Values are means ± standard deviations (SD). For the same species, lower-case letters indicate significant differences (*p* < 0.05) between leaf sides. For the same leaf side, capital letters indicate significant differences (*p* < 0.05) between species.

Species	Leaf Side	*θ_w_* (°)	*θ_g_* (°)	*θ_d_* (°)
Lettuce	Adaxial	75.6 ± 7.4 aC	54.1 ± 6.8 aC	62.4 ± 5.1 aC
	Abaxial	57.7 ± 11.6 bB	58.1 ± 5.8 aB	55.8 ± 5.7 bB
Broccoli	Adaxial	131.3 ± 5.5 bB	138.0 ± 5.0 aA	97.9 ± 3.6 bB
	Abaxial	137.9 ± 6.7 aA	136.6 ± 3.7 aA	105.4 ± 6.4 aA
Leek	Adaxial	143.5 ± 5.1 aA	133.7 ± 5.4 bB	108.4 ± 4.9 aA
	Abaxial	143.4 ± 3.2 aA	139.4 ± 4.0 aA	103.4 ± 6.8 bA

**Table 3 plants-12-02357-t003:** Total surface free energy (*γ*_s_), with the contribution of the Lifshitz–van der Waals c (*γ*^LW^) and acid-base (*γ*^AB^) components, polarity (*γ*^AB^
*γ*^−1^) and solubility parameter (*δ*) of the adaxial and abaxial leaf surfaces of lettuce, broccoli and leek plants.

Species	Leaf Side	*γ*^LW^ (mJ m^−2^)	*γ*^AB^ (mJ m^−2^)	*γ*_s_ (mJ m^−2^)	*Polarity* (*γ*^AB^ *γ*^s−1^, %*)*	*δ* (MJ^1/2^ m^−3/2^)
Lettuce	Adaxial	27.2	8.7	35.9	24.3	18.2
	Abaxial	31.0	8.6	39.6	21.7	19.6
Broccoli	Adaxial	9.4	3.8	13.3	28.9	8.6
	Abaxial	6.9	0.8	7.6	10.3	5.7
Leek	Adaxial	5.9	0.9	6.8	12.9	5.2
	Abaxial	7.5	0.1	7.6	0.8	5.7

## Data Availability

The data presented in this study are available on request from the corresponding authors.

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
