# Peer review of "Evaluating Leaf Wettability and Salt Hygroscopicity as Drivers for Foliar Absorption"

_plants, 2023, doi:10.3390/plants12122357_

Round 1
Reviewer 1 Report
Reviewer report for manuscript entitled "Evaluating leaf wettability and salt hygroscopicity as drivers for foliar absorption". This paper addresses the evaluation of magnesium salt absorption rate at different deliquescence and efflorescence relative humidity values. The experimental work is well designed and the work is well structured. The conclusion is clear which provided added value to both scientific and plantation industries, where leaf wettability govern the foliate magnesium absorption and can be considered as the main factor in this process. However, I recommend to do the following minor changes before acceptance f this work as follows:
- In the introduction part, need to add more review about factors affecting leaf wettability and how different factors (biological and environmental factors) affect leaf wettability.
- In Materials and methods part This part is well written and provided all necessary information to understand the experimental set up, process for experiment, and analysis.
- The result part is well-structured. However, some figures are very poor in resolution as follows:
- Figure 1, its very poor in resolution (the X axe is not readable) and need to provide clearer version.
- Figure 2, for SEM, this figure is very poor in quality, non-readable and the magnification bar is hard to see. This MUST be changed
- Other figures are well presented and recommend to remove the brackets of the Y axe in figure 6 for example [chloride] need to be just chloride and keep the bracket for the units ONLY.
- The discussion section is well written and provided a deep Interpretation of data and acceptable discussion.
- Conclusion part need to be revised and to provide a strong in depth recommendation for the application of the data produced. In addition, need to provide in which direction further research can be carried out.
- In general the work is well written and worthy for publication after minor revision.
Author Response
Dear Reviewer 1,
Thank you very much for your comments. We carried efforts for revising the entire manuscript as suggested.
We followed your recommendations as described below and also realised that the SD bars had moved in anion concentration Figures which have been also corrected.
Below, you can find a response to your comments as follows:
Reviewer 1 report for manuscript entitled "Evaluating leaf wettability and salt hygroscopicity as drivers for foliar absorption".
This paper addresses the evaluation of magnesium salt absorption rate at different deliquescence and efflorescence relative humidity values. The experimental work is well designed and the work is well structured. The conclusion is clear which provided added value to both scientific and plantation industries, where leaf wettability govern the foliate magnesium absorption and can be considered as the main factor in this process. However, I recommend to do the following minor changes before acceptance f this work as follows:
- In the introduction part, need to add more review about factors affecting leaf wettability and how different factors (biological and environmental factors) affect leaf wettability.
This has been added as suggested, thank you.
- In Materials and methods part This part is well written and provided all necessary information to understand the experimental set up, process for experiment, and analysis.
- The result part is well-structured. However, some figures are very poor in resolution as follows:
Thank you very much for your comments. We revised the Figures as suggested, including also the SD in the anions Figure which were moved.
- Figure 1, its very poor in resolution (the X axe is not readable) and need to provide clearer version.
This has been modified as suggested
- Figure 2, for SEM, this figure is very poor in quality, non-readable and the magnification bar is hard to see. This MUST be changed
This has been modified as suggested
- Other figures are well presented and recommend to remove the brackets of the Y axe in figure 6 for example [chloride] need to be just chloride and keep the bracket for the units ONLY.
Dear Reviewer, brackets indicate concentrations by general convention.
Please, see e.g., https://en.wikipedia.org/wiki/Defining_equation_(physical_chemistry), as an example.
Parentheses are used for the units (not brackets which indicate concentration).
We hence left the mineral element concentrations in Y axes indicated in brackets, as generally expressed. Thank you for your understanding but otherwise the Y-axis legends would become too long!
- The discussion section is well written and provided a deep Interpretation of data and acceptable discussion.
Thank you very much for your comments.
- Conclusion part need to be revised and to provide a strong in depth recommendation for the application of the data produced. In addition, need to provide in which direction further research can be carried out.
Thank you very much for your comments. We modified the conclusions as suggested.
- In general the work is well written and worthy for publication after minor revision.
Thank you very much for your comments.
Reviewer 2 Report
Thank you for your invitation to review the review article titled “Evaluating leaf wettability and salt hygroscopicity as drivers for foliar absorption”. The article is interesting since it discusses scientific literature indicating some limitations of the traditional plant nutrition methods. The article is clear in the display of backgrounds as well as in results. Moreover, it also shows a relevant literature research even if the only 11 on 66 cited articles have been published in the last 5 years. A revision of language is also recommended.
Introduction: clear statement and it serves the research topic
Materials and Methods: clear statement
Results: very good statement
Discussion: poor section and must be more depth and improved
Moderate editing of English language required
Author Response
Dear Reviewer,
thank you very much for your comments.
We carried efforts for revising the entire manuscript including the English language, text coherence and significance.
We did our best to revise the Discussion and tried to make sure it makes sense, also in relation to the existing literature. We added and introductory paragraph for making clear the significance of our investigation, and also clarifying the meaning of our study for readers.
We now made the conclusions more straight forward, providing now some clear recommendations and clear suggestions for future studies.
We hope you find these improvements sufficient, provided also the suggestions of Reviewer 1.
Thank you very much for helping us improve our draft.